# Variations of the magnetic declination at mid-latitude European stations during the Carrington-like event on 29 October 2003

Fridrich Valach[1], Magdaléna Váczyová[1], and Eduard Koči[1,2]

[1]Geomagnetic Observatory, Earth Science Institute, Slovak Academy of Sciences, Komárňanská 108, 947 01 Hurbanovo, Slovakia
[2]Slovak Central Observatory, Komárňanská 137, 947 01 Hurbanovo, Slovakia

**Correspondence:** F. Valach (geoffval@savba.sk)

**Abstract.** Based on the declination observed at mid-latitude European stations (geomagnetic latitudes 34°N–58°N), we studied the current system that is a candidate for the cause of the sharp drop in horizontal intensity ($H$) of the geomagnetic field that occurred in that part of the globe at ∼9:00 of magnetic local time on 29 October 2003. The newest knowledge says that the current system consisted of a pair of field-aligned currents (FACs) forming a dayside current wedge: in the early afternoon

sector, it was a stationary upward FAC, and in the dawn sector, it was a westward-moving downward FAC. Simultaneously with the drop in $H$, the current wedge caused a sine-like profile in declination observed at the mid-latitude European stations. By studying this profile at individual observatories, we found the velocity of $-1.08$ °/min $\pm 0.38$ °/min for the wedge centre, roughly half the velocity of the westward-moving downward FAC. Our results contribute to arguments that the dayside current wedge was the probable cause of the $H$-drop on 29 October 2003.

**1  Introduction**

The sharp variation of the horizontal intensity ($H$) of the geomagnetic field observed at mid-latitude European stations on 29 October 2003 between 06:30 and 07:30 UT (with a minimum at 06:58 UT, corresponding to ∼09:00 of magnetic local time, MLT, in this part of the world) has recently aroused renewed interest (e.g. Cid et al., 2015; Ohtani, 2022; Love and Mursula, 2024). This event was part of the Halloween storm, one of the most intense magnetic storms in the era of digital observatories.

Moreover, since Cid et al. (2015) pointed out the possible similarity of the mechanism in the $H$ variation of this event (they named it Carrington-like event C03) to that of the famous Carrington event of 2 September 1859 (abbrev. C59), the study of the 29 October 2003 event, or C03, has gained importance.

    The C59 event was initially assumed, since its rediscovery by Tsurutani et al. (2003), to be a magnetic storm caused by the suddenly enhanced symmetric ring current (see, e.g., Tsurutani et al., 2003). However, the study of the Burton equation by

Love and Mursula (2024) has provided convincing arguments that this mechanism may not be the correct explanation for the C59 event and that the event was significantly influenced by the partial ring current or by a dayside field-aligned current (FAC) or ionospheric current. The possible important role of other magnetospheric currents besides the symmetric ring current has also been suggested by others (e.g. Siscoe et al., 2006).

Cid et al. (2015) noted the similarity of the temporal profiles of the $H$-spike variation at Colaba, India, and the European observatory Tihany (IAGA code THY) in the C59 and C03 events, respectively. They also expressed their belief, supported by the study of the $H$-variations at mid-latitudes in different parts of the Earth on 29 October 2003, that the cause of the $H$-spike was not the ring current but that FACs played a major role in C03 and probably also in C59.

Ohtani (2022) made the idea of the FACs that caused the $H$ variation of C03 and possibly also C59 more concrete (see Fig. 6b in Ohtani, 2022). It was a dayside current wedge related to the solar wind interaction with the front part of the magnetosphere. One part of the wedge was an upward FAC in the sector shortly after magnetic noon; this FAC held its position. In the morning sector, the wedge consisted of a downward FAC; during the $H$-spike occurrence, this FAC was moving from the early morning sector through the dawn towards the late-night sector. The current system was closed primarily via the nightside auroral-oval ionosphere. The model of Ohtani (2022) explains both the $H$ and $D$ (declination) variations in C03.

Besides the rapid $H$ variations, the mid-latitude European observatories also recorded interesting $D$ variations. To our knowledge, these variations have not yet been studied in detail. We focus this study on them. Ohtani's idea of a dayside current wedge implies that the centre of the wedge (located in the morning sector), where the effects on the declination of the motionless upward FAC and the westward-moving downward FAC are balanced, must be moving westward. In this paper, we aim to contribute to confirming the idea of Ohtani (2022) about the mechanism of the C03 event, and thus indirectly also C59, by tracing the motion of the current wedge centre.

## 2  Method and data

The idea about tracing the centre of the dayside wedge current system during the C03 event is sketched in Fig. 1. Along with the westward movement of the place where the downward FAC connects the ionosphere, the centre of the wedge also shifts westward. In the idealised case, the speed of the downward FAC movement is twice the velocity of the wedge centre.

To explain the variation in declination, consider a mid-latitude magnetic station, which is initially to the west of the imagined centre of the wedge. As the current wedge starts to be fed by a pair of FACs, a negative declination variation arises at that station and deepens as the electric current in the system becomes stronger; this happens together with deepening the $H$ depression. As the downward FAC moves away and the centre of the wedge moves closer to the station, the influence on the declination from the downward FAC is balanced by the opposing influence from the upward FAC. When the station is in the centre of the current wedge, the declination variation should disappear. Consequently, as the centre of the wedge gradually moves further west, the influence of the upward FAC gradually increases, causing positive $D$ variation. The positive $D$ variation eventually decreases and disappears when the electric current stops flowing through the wedge; this happens when the horizontal intensity depression at the station weakens and disappears.

The instant of passing the centre of the current wedge through the magnetic station can be identified as the moment when the variation of $D$ was zero ($\Delta D = 0$), which took place when the variation was changing from its negative to positive excursions. Based on the time points of passages of the wedge centre and the geomagnetic longitudes of the observatories, we can straightforwardly determine the speed of the motion of the wedge centre.

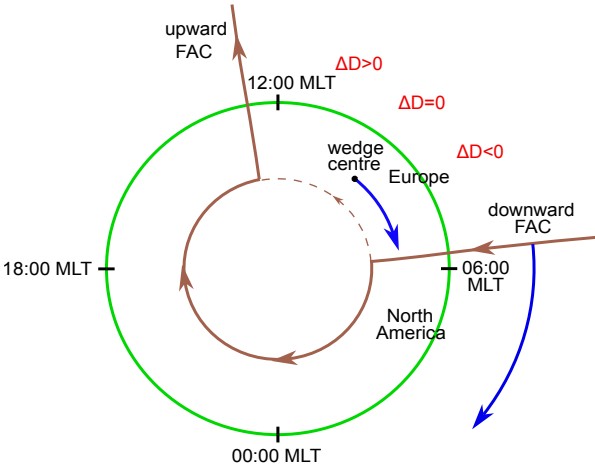

**Figure 1.** A sketch of the dayside current wedge during the C03 event adapted from (Ohtani, 2022). The upward FAC is static, while the downward FAC is moving westward. The upward FAC produces a positive $D$ variation ($\Delta D > 0$) in the mid-latitudes, and the downward one induces a negative $\Delta D$. In the centre of the wedge, the influence of the upward and downward FACs is balanced ($\Delta D = 0$). The green circle indicates the line of a geomagnetic mid-latitude. In the brown colour, we drew the set of two FACs (upward and downward) and the night-side closure across the auroral oval. The thin brown dashed line shows the less important closure over the day side. Indicated is the approximate location of Europe and North America during event C03.

The observatory data used in our study (Table 1) were obtained from the INTERMAGNET database. The geomagnetic positions of the observatories were corrected geomagnetic (CGM) coordinates, which were calculated for the year 2003 using the (Papitashvili and Papitashvili, n.d.) calculator.

Since we were determining the movement of the wedge centre relative to the geocentric solar magnetospheric (GSM) coordinate system, we had to consider that the observatories moved together with the rotating planet eastwards at approximately 0.25 °/min.

To obtain the variations of the geomagnetic elements $H$ and $D$ (Fig. 2), we subtracted undisturbed values from the time series of one-minute averages of horizontal intensity and declination. We determined these undisturbed values for each observatory as the averages of the values between 06:00 UT and 07:59 UT on 12 October 2003 (at that time, the $Kp$ index was $0+$) and 8 November 2003 (the $Kp$ index was $1-$). The two-hour interval between 06:00 UT and 07:59 UT was chosen because it represents the same part of the days as that during which the C03 event occurred. The days 12 October and 8 November were picked out because they were the quietest days close to 29 October. Here, the geomagnetic activity was measured with the $Kp$ index in that two-hour interval. We obtained the $Kp$ index values from the GFZ Potsdam webpage (Matzka et al., 2021a, b).

In the end, the speed of the wedge centre can be compared with the speed of movement of the place where the downward FAC connects to the ionosphere. The latter speed was obtained by the procedure described in (Ohtani, 2022, Appendix A) by using the minima of the $D$ variations and the zero value of the $H$ variations at observatories OTT and STJ.

**Table 1.** Geographic and geomagnetic (CGM) coordinates of the observatories used in the study.

| IAGA code | Geographic | | CGM | |
|---|---|---|---|---|
| | Latitude | Longitude | Latitude | Longitude |
| AQU | 42.38°N | 13.32°E | 34.23°N | 87.31°E |
| BDV | 49.08°N | 14.01°E | 44.44°N | 89.35°E |
| BEL | 51.84°N | 20.79°E | 47.60°N | 95.96°E |
| CLF | 48.02°N | 2.27°E | 43.37°N | 79.21°E |
| DOU | 50.10°N | 4.60°E | 45.86°N | 81.70°E |
| EBR | 40.82°N | 0.49°E | 33.93°N | 76.10°E |
| FUR | 48.16°N | 11.28°E | 43.35°N | 86.78°E |
| HLP | 54.61°N | 18.82°E | 50.72°N | 95.04°E |
| HRB | 47.87°N | 18.19°E | 43.04°N | 92.71°E |
| ISK | 41.06°N | 29.06°E | 35.56°N | 101.55°E |
| LER | 60.10°N | 358.80°E | 57.92°N | 80.82°E |
| LOV | 59.34°N | 17.82°E | 55.93°N | 95.88°E |
| MAB | 50.30°N | 5.68°E | 46.06°N | 82.66°E |
| NCK | 47.63°N | 16.72°E | 42.73°N | 91.36°E |
| NGK | 52.07°N | 12.67°E | 47.96°N | 89.03°E |
| NUR | 60.51°N | 24.65°E | 56.96°N | 102.09°E |
| OTT | 45.40°N | 284.45°E | 55.74°N | 1.32°E |
| STJ | 47.59°N | 307.32°E | 53.34°N | 31.29°E |
| SUA | 44.68°N | 26.12°E | 39.57°N | 99.27°E |
| THY | 46.90°N | 17.89°E | 41.89°N | 92.23°E |
| UPS | 59.90°N | 17.35°E | 56.56°N | 95.73°E |
| WNG | 53.74°N | 9.07°E | 50.01°N | 86.54°E |

## 3 Results and discussion

The moments when the centre of the wedge successively passed through each station (see Fig. 3, which is a detail from Fig. 2b) and the magnetic longitudes (Table 1) of the stations were used to determine the slope of the line in Fig. 4. We obtained the value $-1.33$ °/min $\pm$ 0.38 °/min. After subtracting the motion of the stations together with the rotating planet, we determined the angular velocity of the motion of the current wedge centre to be $-1.08$ °/min $\pm$ 0.38 °/min, with the negative sign indicating that the motion is in the direction opposite to the geomagnetic longitude growth. The accuracy of the result is expressed as the $3\sigma$ error.

To determine the velocity of the movement of the downward FAC, we estimated from Fig. 5 the moments when the FAC passed through the regions that lie polewards of two observatories on the American continent, namely STJ and OTT (see

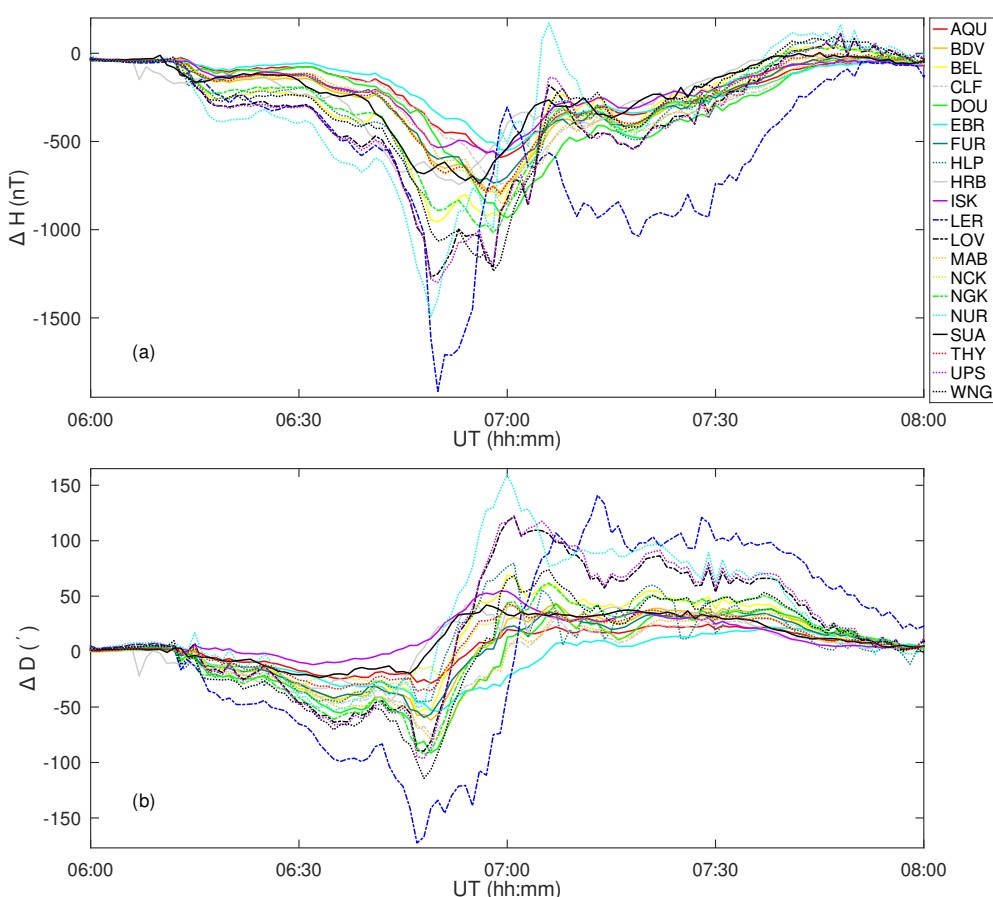

**Figure 2.** Variations of the (a) horizontal intensity and (b) declination at mid-latitude European observatories during event C03 on 29 October 2003.

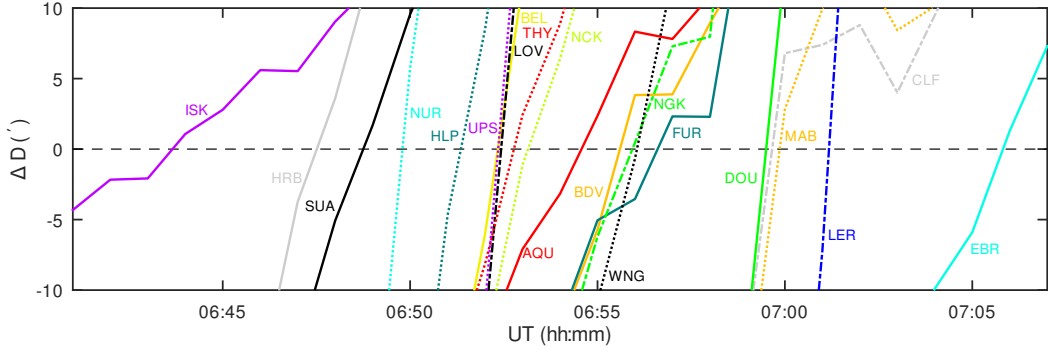

**Figure 3.** Variations in declination at mid-latitude European observatories when the centre of the current wedge passed through them.

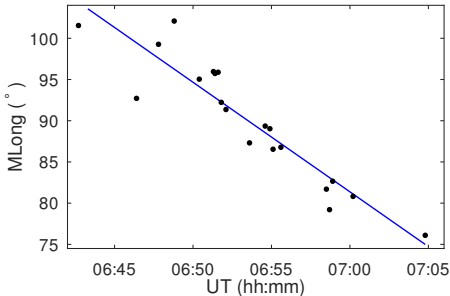

**Figure 4.** Plot of the geomagnetic longitudes of the stations against the moments of the current wedge passages through the stations.

Appendix A in Ohtani, 2022). The downward FAC appeared polewards from these stations when the sharp drop in $H$ was observed in Europe. Because we could not determine the moments from Fig. 5 unambiguously in most cases, we had to consider a visually estimated error ($\sim 3\sigma$) in determining these values. For the OTT observatory, we determined the transit of FAC based on the declination and horizontal intensity as 07:28 UT $\pm$ 1 min and 07:23.7 UT $\pm$ 1.5 min, respectively. For STJ, we determined the FAC transit from declination as 07:21.5 UT $\pm$ 2.5 min; from horizontal intensity, we determined the precise moment at 07:08.5 UT. After subtracting the motion of the observatories together with the rotating planet (note that the current system associated with the pair of FACs is not subject to this rotational motion), we found the rate of motion of the point at which the downward pointing FAC feeds the ionospheric layer of the auroral oval. The value obtained is $-2.51$ °/min $\pm$ 0.39 °/min, with the negative sign indicating that the motion is opposite to the direction of geomagnetic longitude growth.

Taking into account also the uncertainties in the determined values, the velocity of of the downward FAC motion is roughly twice the velocity of the motion of the dayside current wedge centre, which is consistent with the prediction of the model of Ohtani (2022). With this result, we contribute to validating Ohtani's hypothesis about the cause of the C03 event and, indirectly, about the cause of the Carrington event C59, too.

## 4 Conclusions

In this paper, we highlighted the interesting declination variations in the mid-latitude part of Europe during the sharp and deep spike-like excursion of horizontal intensity in this region during the Carrington-like event C03 on 29 October 2003. We assumed the existence of a current wedge fed by a pair of FACs as presented by Ohtani (2022). Based on the declination profiles at the mid-latitude European observatories, we determined the velocity of motion of the wedge centre to be $-1.08$ °/min $\pm$ 0.38 °/min, which is approximately half the velocity of motion of the point where the downward FAC feeds the ionospheric auroral oval. The result is consistent with the model of Ohtani (2022).

*Author contributions.* FV devised the study idea and methodology, analysed the data, and wrote most of the manuscript. MV and EK verified the results and contributed to writing the manuscript.

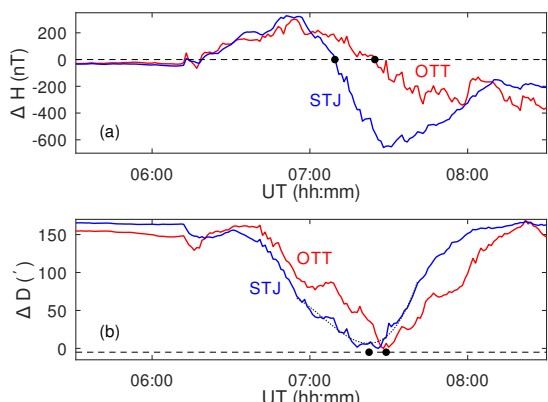

**Figure 5.** Variations in (a) horizontal intensity and (b) declination at observatories OTT and STJ. The curves for horizontal intensity are lowered by the non-disturbed values determined from nearby days. The curves for the declination variations are shifted vertically so that the minima are next to each other for easier visual comparison. The black dots indicate the moments of the FAC transits. The thin dotted black line in panel (b) fits a third-degree polynomial to the STJ declination series at a time around the minimum, which occurred at ∼07:21.5 UT.

*Competing interests.* The authors declare that they have no conflict of interest.

*Acknowledgements.* The results presented in this paper rely on data collected at magnetic observatories. We thank the national institutes that support them and INTERMAGNET for promoting high standards of magnetic observatory practice (www.intermagnet.org). The research presented in this paper was funded by the Scientific Grant of the Ministry of Education of Slovak Republic and the Slovak Academy of Sciences, grant VEGA no. 2/0003/24.

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
