# Peer review of "Variations of the magnetic declination at mid-latitude European stations during the Carrington-like event on 29 October 2003"

_EGUsphere, 2025_

## Author Comment (AC1)

**Replies to the Reviewer #1's comments**

The authors have provided a concise study about the declination variations during the major Halloween storm in 2003. The observed declination changes are consistent with the dayside current wedge model presented by Ohtani (2022). As such, the paper provides a valuable contribution to confirming the model. I can recommend the publication of the manuscript after some minor revisions.

*Reply:*

*Dear Reviewer, Thank you very much for the time and effort you have put into our manuscript. We agree with all your comments and will incorporate them in the new version of the manuscript.*

**Abstract:** Presently the text is very short. A few more words should be spent, describing the H drop during the storm, e.g. at which latitude is it observed, which local time? What is the importance of the H drop for your interpretation of the D components?

*Reply:*

*We will add the suggested information to the abstract text.*

*The new abstract will read as follows (the new parts of the text are written in red):*

*'Based on the declination observed at mid-latitude European stations (geomagnetic latitudes 34°N–58°N), we studied the current system that is a candidate for the cause of the sharp drop in horizontal intensity (H) of the geomagnetic field that occurred in that part of the globe at ∼9:00 of magnetic local time on 29 October 2003. The newest knowledge says that the current system consisted of a pair of field-aligned currents (FACs) forming a dayside current wedge: in the early afternoon sector, it was a stationary upward FAC, and in the dawn sector, it was a westward-moving downward FAC. Simultaneously with the drop in H, the current wedge caused a sine-like profile in declination observed at the mid-latitude European stations. By studying this profile at individual observatories, we found the velocity of −1.08 °/min ± 0.38 °/min for the wedge centre, roughly half the velocity of the westward-moving downward FAC. Our results contribute to arguments that the dayside current wedge was the probable cause of the H-drop on 29 October 2003.'*

*We have also added information about magnetic local time in Introduction (1st sentence):*

*'The sharp variation of the horizontal intensity (H) of the geomagnetic field observed at mid-latitude European stations on 29 October 2003 between 06:30 and 07:30 UT (with a minimum at 06:58 UT, corresponding to ∼09:00 of magnetic local time, MLT, in this part of the world) has recently aroused renewed interest (e.g. Cid et al., 2015; Ohtani, 2022; Love and Mursula, 2024).'*

**Fig. 2:** This is a very busy plot with many curves. For improvement it should be enlarged and the presented time span truncated to 06 to 08 UT. This will improve the readability a little.

*Reply:*

*We will modify Figure 2 according to your recommendation. The revised Fig. 2 will be larger than the previous one and will look like this:*

[Figure]

**Fig. 5**: The time for the dot, marking the ΔD minimum at STJ, seems to be very arbitrary. Applying a smoothing to the curves may help to come to more justified results.

*Reply: In Figure 5, we will add a fitting curve for the part of the time series around the minimum of the declination at STJ.*

*We will use a 3rd-degree polynomial for this purpose. We will also add a new sentence to the figure caption (the last sentence in the caption), which will read as follows: 'The thin dotted black line in panel (b) fits a third-degree polynomial to the STJ declination series at a time around the minimum, which occurred at ∼07:21.5 UT.'*

*In the manuscript, we do not provide the detailed procedure by which we arrived at this curve. But the following 'technical' information might be of the reviewer's interest:*

*We tried fitting with polynomials of degrees 2, 3, and 4. We took data from the interval 06:54 to 07:40 UT. From the 2nd-degree polynomial, we obtained a minimum at 7:18 UT (on sight, the fitted curve did not represent the time series well). From the 3rd-degree polynomial, we obtained a minimum between 7:21 and 7:22 UT (exactly 7:21.24 UT), the same time as we estimated by eye, i.e. 7:21.5 UT. From the 4th-degree polynomial, we obtained the same value as from the 3rd-degree polynomial.*

*We also compared the above results with the results of smoothing by the moving medians: the time window over 3, 5, 7, 9, and 11 minutes yielded minima at 7:24.5 UT, 7:25 UT, 7:24.5 UT, 7:21 UT, and 7:20 UT, respectively. The curve has two competing minima which is why these values are so different from each other.*

*We consider the best estimate of the ΔD-minimum time to be that yielded by the fit by a polynomial of degree 3 and the estimation by eye (with the same result, i.e., 7:21.5 UT).*

*The revised Fig. 5 will be as follows:*

[Figure]